# Spiritual well-being as a protective factor for endothelial dysfunction in clinically healthy adults

Andre Casarsa[1,2☯‡], Pedro Bastos de Medeiros[1,3☯], Julio Cesar Tolentino[1,2‡], Isadora de Sa Guimaraes[1], Kelen Carolina Silva Cruz[1], Leticia Silva Flor dos Santos[1], Matheus Nakazato Tinoco[1], Maria de Fatima Martins Gil Dias[2], Ana Lucia Taboada Gjorup[1,2], Sergio Luis Schmidt[1,2*‡]

**1** Department of Internal Medicine, University Hospital Gaffrée Guinle, Federal University of the State of Rio de Janeiro, Rio de Janeiro, Rio de Janeiro, Brazil, **2** Postgraduate Neurology Department, Federal University of the State of Rio de Janeiro (UNIRIO), Rio de Janeiro, Rio de Janeiro, Brazil, **3** Department of Internal Medicine, University Hospital Clementino Fraga Filho, Medical School, Federal University of Rio de Janeiro, Rio de Janeiro, Rio de Janeiro, Brazil

☯ These authors contributed equally to this work.
‡ These authors share first authorship on this work.
* slschmidt@terra.com.br

## Abstract

### Background

Endothelial dysfunction (ED) is an early marker of cardiovascular disease (CVD), influenced by both physiological and psychosocial factors. While depression and anxiety are known contributors to ED, the role of spiritual well-being (SWB) in vascular health has been relatively less explored in the literature.

### Objective

To investigate the association between SWB and ED in clinically healthy adults, controlling for mental health variables and conventional cardiovascular risk factors.

### Methods

In this cross-sectional study, 148 individuals aged 18–60 years were assessed using validated instruments: FACIT-Sp for SWB, PHQ-9 for depression, GAD-7 for anxiety, and brachial artery flow-mediated dilation (FMD) for endothelial function. Logistic regression and discriminant analyses were performed to identify independent predictors of ED and the spiritual dimensions most associated with vascular health.

### Results

ED was identified in 39.2% of participants. Multivariate logistic regression indicated that SWB (OR = 0.929; $p = 0.005$), body mass index (OR = 1.130; $p = 0.016$), generalized anxiety disorder (OR = 2.551; $p = 0.035$), and major depressive episode (OR

**Data availability statement:** All relevant data are within the paper.

**Funding:** The author(s) received no specific funding for this work.

**Competing interests:** The authors have declared that no competing interests exist.

= 3.740; $p = 0.038$), were significantly associated with ED. Among these, SWB was significantly inversely associated with ED even after excluding participants with anxiety or depression. Discriminant analysis further indicated that inner peace and life purpose—but not faith—significantly distinguished individuals with and without ED.

## Conclusion

SWB, particularly dimensions related to inner peace and meaning, is independently associated with preserved endothelial function in healthy adults. These findings support the inclusion of psychosocial and spiritual dimensions in cardiovascular risk assessment and prevention strategies.

## Introduction

The endothelium is a critical component of the vascular system, acting as a single layer of cells lining blood vessels, and plays a vital role in maintaining vascular homeostasis [1]. Endothelial dysfunction (ED), characterized by impaired vasodilation and pro-inflammatory and pro-thrombotic states, represents an early and pivotal event in the pathogenesis of atherosclerosis. As such, ED serves as a key biomarker of cardiovascular risk, preceding overt clinical manifestations of cardiovascular disease (CVD) [2].

While several psychosocial factors—such as anxiety and depression—are recognized as contributors to ED, emerging evidence highlights the potential of spirituality as a protective element within this context [3,4]. Beyond its well-established role in strengthening psychological resilience and alleviating symptoms of anxiety and depression, spirituality may exert beneficial effects on vascular health through both direct physiological mechanisms and indirect modulation of stress responses [5]. As such, integrating spirituality into cardiovascular risk assessment frameworks may provide a more comprehensive and multidimensional understanding of the psychosomatic determinants of endothelial integrity and overall vascular function.

Spirituality is a multidimensional and intrinsic aspect of human experience, characterized by inner peace, harmony, and the search for meaning and purpose, including, when applicable, the faith [6]. Unlike religiosity—which is tied to institutionalized practices and dogma—spirituality encompasses broader subjective experiences that are not necessarily linked to formal religious beliefs [7]. Validated instruments, such as the Functional Assessment of Chronic Illness Therapy–Spiritual Well-being (FACIT-Sp), have been developed to measure spirituality comprehensively by focusing on three core dimensions: peace, purpose, and faith [8,9].

Investigating spirituality as an independent factor could therefore provide a more comprehensive understanding of vascular health determinants in clinically healthy adults.

This study aimed to explore the relationship between mental health variables—namely depression, anxiety, and spirituality—and the development of endothelial dysfunction (ED) in clinically healthy individuals. Specifically, it addressed the following

questions: (1) Was spiritual well-being (SWB) associated with ED? (2) Did this association persist after excluding individuals with anxiety or depressive symptoms? (3) Which SWB dimensions—peace, meaning, or faith—showed the strongest association with ED?

## Materials and methods

This cross-sectional study recruited a convenience sample of clinically healthy individuals aged 18–60 years between January 2022 and October 2024. Participants were enrolled through institutional outreach at a university hospital in Rio de Janeiro, Brazil, using informational flyers, verbal invitations during hospital or academic events, and personal referrals from healthcare professionals. The sample included university students, healthcare professionals, administrative staff, and relatives of patients, reflecting a heterogeneous but health-conscious population.

To ensure clinical health status, all participants underwent a comprehensive screening protocol including medical interviews and physical examinations, as well as ancillary tests—electrocardiogram, carotid Doppler ultrasound, and transthoracic echocardiogram (Fig 1)—to exclude underlying cardiovascular, neurological, or psychiatric conditions. Individuals with any chronic disease (e.g., diabetes mellitus, hypertension, dyslipidemia, thyroid dysfunction), previous psychiatric diagnosis, current use of psychotropic medications, or abnormal cardiovascular findings were excluded. All participants provided written informed consent after receiving a detailed explanation of the study objectives, procedures, potential

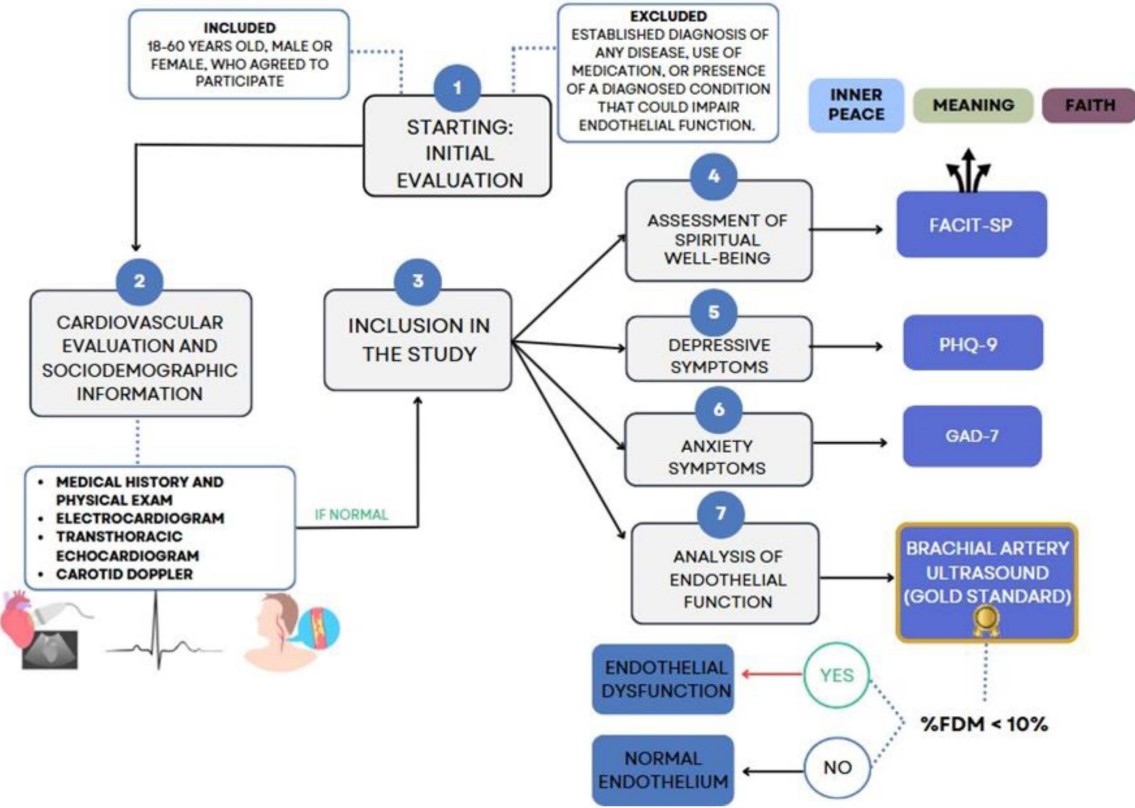

**Fig 1. Flowchart: recruitment, screening, and assessment flow for the cross-sectional study.** Clinically healthy participants aged 18–60 underwent cardiovascular screening and psychosocial evaluations, including assessments of SWB (FACIT-Sp), depressive symptoms (PHQ-9), and anxiety symptoms (GAD-7). Endothelial function was evaluated through brachial artery ultrasound using flow-mediated dilation (FMD), with ED defined as FMD < 10%. Participants were included only if cardiovascular and clinical screening results were within normal limits.

risks, and benefits. The consent process was conducted individually in a private setting, ensuring the opportunity to ask questions before signing. Only adults (≥18 years) were enrolled; therefore, no parental or guardian consent was required. The study protocol, including the consent procedure, was reviewed and approved by the Research Ethics Committee of UNIRIO (CAAE: 30547720.3.0000.0008) in accordance with the Declaration of Helsinki.

The FACIT-Sp scale, a widely recognized instrument for clinical and psychosocial research, culturally adapted and validated for the Brazilian population [10], as well as for several other cultural and linguistic contexts worldwide [5,11], was used to assess SWB. This 12-item questionnaire evaluates three core domains: Peace (items 1, 4, 6, 7), Purpose (items 2, 3, 5, 8), and Faith (items 9–12). Participants rated statements on a 5-point Likert scale (0 = "not at all" to 4 = "very much"), with reverse coding applied to items 4 and 8 to mitigate response bias. Total scores range from 0 to 48, with higher scores indicating greater SWB. Previous validation studies in Brazilian populations have confirmed high internal reliability [10].

The assessment of depressive symptoms was conducted through the Patient Health Questionnaire-9 (PHQ-9), a widely recognized instrument developed by Kroenke et al. [12] to measure both the occurrence and intensity of depression. This tool aligns with the diagnostic criteria for major depressive episodes outlined in the DSM and employs a four-point Likert-type scale ranging from 0 ("never") to 3 ("nearly daily"). Participants rate how frequently they experienced symptoms over the preceding two weeks, resulting in a total score between 0 and 27. Higher scores reflect increased depressive severity. A positive screen for depression required participants to endorse five or more symptoms occurring on "more than half the days," with at least one being a core symptom (persistent low mood or loss of interest/pleasure). Notably, any indication of suicidal ideation or self-harm (item nine) was automatically classified as a depressive symptom, regardless of its reported frequency. This approach aligns with established diagnostic guidelines [13].

Generalized Anxiety Disorder (GAD) was assessed using the Generalized Anxiety Disorder 7-item scale (GAD-7) [14], one of the most reliable and widely used screening tools in clinical and research settings, validated for the Brazilian population [14]. The questionnaire consists of seven items rated on a 4-point Likert scale: 0 ("not at all"), 1 ("several days"), 2 ("more than half the days"), and 3 ("nearly every day"). Total scores range from 0 to 21, with higher scores reflecting greater anxiety severity. A score above 10 points on the GAD-7 indicates a positive screening for generalized anxiety disorder. The scale's design allows for a quantitative evaluation of symptom intensity, where elevated scores correlate with more pronounced anxiety-related impairments [14,15].

In the present study, internal consistency reliability coefficients were high for the FACIT-Sp (Cronbach's alpha [α] = 0.862), PHQ-9 (α = 0.814), and GAD-7 (α = 0.890).

Endothelial function was assessed as a continuous variable using measurements of the brachial artery's flow-mediated dilation (FMD), following standard protocol guidelines [16]. Examinations were conducted by a single observer in a controlled environment at 22–24°C. Participants fasted for at least four hours, lay supine, and rested for 10 minutes before testing. Monitoring included three cutaneous ECG electrodes, with blood pressure measured on the left arm. The basal diameter of the brachial artery was calculated as the mean of 3 pre-hyperemic measurements. Then, the brachial artery was occluded with a pressure cuff at 200 mmHg for five minutes. After unclamping the artery, three new measurements of the diameter of the brachial artery were performed between 45–60 seconds post-occlusion. These were used to calculate the mean post-hyperemic diameter. A subject was considered as presenting ED, when FMD < 10% which indicates a post-hyperemic brachial artery dilation < 10%. The echocardiographer was blinded to all other clinical and psychiatric assessments. FMD% was calculated as:

$$\textbf{FMD\%} = \textbf{100} \times ((\textbf{Post} - \textbf{hyperemia diameter} – \textbf{Baseline diameter}) / \textbf{Baseline diameter})$$

The presence of ED was defined as FMD < 10%, in line with previous international studies [17,18] and Brazilian cohort data [19,20]. It is important to note, however, that this threshold may be influenced by individual and population-specific characteristics [21].

## Statistical analysis

Descriptive statistics were used to summarize sample characteristics, including demographic, anthropometric, psychological, and vascular parameters. Continuous variables were reported as means and standard deviations (SD), while categorical variables were presented as frequencies and percentages. Spiritual well-being was analyzed using the total score from the FACIT-Sp scale, treated as a continuous variable in all inferential analyses.

To identify independent predictors of ED, a multivariate logistic regression analysis was conducted. The dependent variable was the presence or absence of ED, while the independent variables included age, sex, body mass index (BMI), physical activity level, positive screening for generalized anxiety disorder (GAD-7), SWB, and positive screening for major depressive episode (PHQ-9). For each predictor, odds ratios (ORs) with corresponding 95% confidence intervals (CIs) were estimated. Model fit was evaluated using the chi-square goodness-of-fit test, explanatory power was quantified using Nagelkerke's $R^2$, and overall model performance was assessed by its classification accuracy.

Additionally, another logistic regression model was performed after excluding participants with positive screenings for anxiety or depression, with the aim of evaluating whether spirituality would remain an independent predictor of ED even after excluding individuals with mental health disorders. In this model, only the variables that were significant in the initial analysis (BMI and SWB) were included as predictors, while the dependent variable remained the presence or absence of ED. The model's overall significance, the proportion of variance explained (Nagelkerke's $R^2$), and its classification accuracy were also assessed.

Discriminant analysis was conducted to examine which variables best distinguish individuals with and without ED, considering spiritual dimensions such as Peace, Meaning in Life, and Faith as predictors. Initially, the equality of group means was tested using Wilk's $\lambda$.

The assumptions underlying discriminant analysis were assessed, including linearity, normality, multicollinearity, homogeneity of variances, and multivariate normal distribution of the predictors. Box's M test was performed to evaluate the homogeneity of covariance matrices. It is worth noting that discriminant analysis is considered robust to moderate violations of this assumption, particularly in the absence of significant outliers. Box's M test results were interpreted alongside inspection of the log determinants to ensure the adequacy of the analysis.

Following the verification of assumptions, the canonical discriminant function was analyzed to determine the strength of association between the predictor variables and group classification, as well as to assess the contribution of each variable to group separation through the structure matrix and group centroids.

## Ethical considerations

The study protocol was approved by the HUGG/UNIRIO Ethics Committee (CAAE: 50323221.2.0000.5258; 30/09/2021) in accordance with the Declaration of Helsinki. Participants provided written informed consent, ensuring confidentiality and voluntary participation. Data collection commenced post-approval, with sufficient methodological detail provided to enable independent replication.

Participants were invited during routine visits or institutional activities. The study aims and procedures were explained verbally, and written informed consent was obtained prior to enrollment. Participants retained the right to withdraw at any time without compromising their access to healthcare services. They were left free to ask questions and to obtain explanations.

## Results

### Sample characteristics and assessed parameters

The study sample (n = 148) exhibited a heterogeneous age distribution, ranging from 19 to 60 years (mean = 30.5 years; SD = 10.9), with a predominance of male participants (55.4%; n = 82). The mean BMI was 25.5 kg/m² (SD = 4.1), classified

as overweight according to World Health Organization (WHO) criteria. The total score of the FACIT-Sp ranged from 9 to 48 points (mean = 32.2; SD = 9.3). Regarding anxiety, as assessed by the GAD-7, scores ranged from 0 to 21 (mean = 7.3; SD = 5.3), and 32.4% screened positive for the disorder. For major depressive episodes, evaluated via the PHQ-9, scores ranged from 0 to 25 (mean = 7.7; SD = 5.3), with 13.5% meeting diagnostic criteria for depression. Regarding physical activity, 57.4% of participants (n = 85) met the minimum recommendation of 150 minutes per week of moderate-to-vigorous exercise.

In the assessment of endothelial function via FMD, the mean value was 10.6% (SD = 5.34). However, 39.2% of the sample (n = 58) exhibited FMD values below the established cutoff indicative of ED (<10%).

## Predictors of endothelial dysfunction

The analysis identified four variables as significant predictors of ED: SWB (OR = 0.929; 95%CI: 0.882–0.978; $p = 0.005$), BMI (OR = 1.130; 95%CI: 1.023–1.249; $p = 0.016$), positive screening for generalized anxiety disorder (OR = 2.551; 95% CI: 1.070–6.084; $p = 0.035$), and positive screening for major depressive episode (OR = 3.740; 95%CI: 1.075–13.005; $p = 0.038$) (Table 1).

In the logistic regression analysis, the FACIT-Sp total score was used as a continuous predictor of ED. Higher SWB scores were significantly associated with lower odds of ED, independent of sociodemographic and psychological covariates. This finding suggests a protective effect of SWB, supporting its role as a continuous, dimensional construct in health-related outcomes.

The overall model was statistically significant [$\chi^2(7) = 35.388$; $p < 0.001$], explained 28.8% of the variance (Nagelkerke $R^2$), and demonstrated a correct classification rate of 73%. Age ($p = 0.146$), sex ($p = 0.197$), and physical activity ($p = 0.879$) were not statistically significant. Among all predictors, SWB was significantly inversely associated with ED, reinforcing its potential protective role.

Prior to model estimation, multicollinearity diagnostics were performed using Variance Inflation Factor (VIF) and tolerance values for all predictors included in the logistic regression model. All VIF values were below 2.0, and all tolerance values exceeded 0.5, which are well within commonly accepted thresholds (VIF < 5.0 and tolerance > 0.2) for retaining predictors in multivariable models.

## Predictors of endothelial dysfunction in individuals without anxiety or depression

A second logistic regression model was conducted including only participants without positive screenings for anxiety or depression (n = 93), following the exclusion of 55 individuals (35 with anxiety, 7 with depression, and 13 with both). The

**Table 1. Multivariate binary logistic regression identifying predictors of ED in a sample of clinically healthy adults (n = 148).**

| Predictor Variable | β | OR (95% CI) | p-value |
|---|---|---|---|
| Spiritual Well-Being (FACIT-Sp) | −0.074 | 0.929 (0.882–0.978) | 0.005** |
| Body Mass Index (kg/m²) | 0.122 | 1.130 (1.023–1.249) | 0.016* |
| Generalized Anxiety Disorder (GAD-7) | 0.937 | 2.551 (1.070–6.084) | 0.035* |
| Major Depressive Episode (PHQ-9) | 1.319 | 3.740 (1.075–13.005) | 0.038* |
| Sex (Male) | 0.542 | 1.720 (0.754–3.922) | 0.197 |
| Age (Years) | 0.030 | 1.146 (0.990–1.074) | 0.146 |
| Physical Activity (Yes) | −0.062 | 0.939 (0.422–2.092) | 0.879 |

Abbreviations: OR = Odds Ratio; CI = confidence interval; β = unstandardized regression coefficient; p = proof value; *p < 0.05; **p < 0.01.

FACIT-Sp = Functional Assessment of Chronic Illness Therapy–Spiritual Well-being;

GAD-7 = Generalized Anxiety Disorder 7-item scale; PHQ-9 = Patient Health Questionnaire-9.

model retained variables previously identified as significant—BMI and SWB—excluding mental health variables due to the restricted sample (Table 2).

Higher SWB remained a significant protective factor against ED (b = –0.066; OR = 0.936; 95%CI: 0.880–0.996; $p = 0.037$), suggesting an independent association between spirituality-related well-being and vascular health. BMI was not a significant predictor in this subsample (b = 0.101; OR = 1.106; 95%CI: 0.988–1.239; $p = 0.081$).

The overall model was statistically significant [$\chi^2(2) = 6.444$; $p = 0.04$], explaining 9.7% of the variance (Nagelkerke $R^2$) and correctly classifying 75.3% of the cases.

## Dimensions of spirituality

This study aimed to identify the factors that best discriminate individuals with and without ED, incorporating spiritual dimensions (peace, meaning in life, and faith). Tests of equality of group means revealed statistically significant differences for Peace ($p < 0.001$) and Meaning in Life ($p = 0.006$), while Faith ($p = 0.103$) was not statistically significant.

The canonical discriminant function exhibited a canonical correlation of 0.377 and an eigenvalue of 0.166. The structure matrix identified Peace (0.998) as the variable most strongly correlated with the discriminant function, followed by Meaning in Life (0.562) and Faith (0.333). Group centroids confirmed the model's discriminatory power: individuals without ED had a positive mean score (0.325), while those with the condition scored negatively (−0.504). Wilks' Lambda ($\lambda = 0.858$; $p < 0.001$) indicated robust group separation. Additionally, Box's M test ($p = 0.524$) validated the homogeneity of covariance matrices, supporting the adequacy of linear discriminant analysis (Figs 2 and 3).

## Discussion

This study provides novel evidence of an association between SWB and endothelial function in clinically healthy adults. Consistent with prior findings [22,23], we confirmed significant associations between mental health conditions—namely, depression and anxiety—and impaired endothelial function. Higher levels of SWB were associated with a lower likelihood of ED, even after accounting for these psychological variables.

The multivariate analysis identified four variables significantly associated with ED: generalized anxiety disorder, major depressive episodes, BMI, and SWB. Among these, SWB demonstrated an inverse association with ED, with higher SWB levels being associated with a lower likelihood of presenting ED. These findings align with previous research suggesting that spirituality contributes to psychological resilience and reduced allostatic load, possibly through the modulation of stress-related physiological pathways, such as the hypothalamic-pituitary-adrenal axis and inflammatory responses [3,24].

To further explore the independence of this association, we performed a second binary logistic regression excluding individuals with positive screenings for anxiety and depression. SWB remained significantly associated with preserved endothelial function in this restricted sample, indicating that its cardiovascular benefits are not solely mediated by improved mental health. While this strengthens the plausibility of a direct link, the cross-sectional design precludes any causal inference. Therefore, we interpret this association with caution and encourage further prospective research to explore directionality.

**Table 2. Logistic regression model evaluating the association between SWB and ED in a subsample of 93 participants without anxiety or depressive symptoms.**

| Predictor Variable | β | OR (95% CI) | p-value |
|---|---|---|---|
| Spiritual Well-Being (FACIT-Sp) | –0.066 | 0.936 (0.880–0.996) | 0.037* |
| Body Mass Index (kg/m²) | 0.101 | 1.106 (0.988–1.239) | 0.081 |

Abbreviations: OR = Odds Ratio; CI = confidence interval; p = proof value; *p < 0.05.

FACIT-Sp = Functional Assessment of Chronic Illness Therapy–Spiritual Well-being.

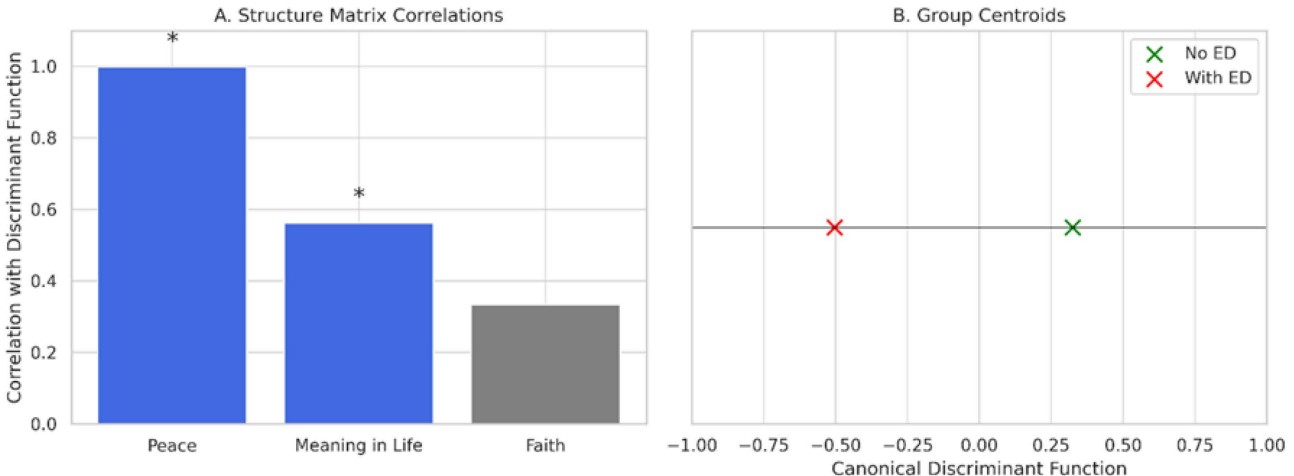

**Fig 2. Discriminant analysis of spiritual dimensions in relation to ED. (A) Structure matrix correlations between each spiritual dimension (Peace, Meaning in Life, and Faith) and the canonical discriminant function.** Inner peace demonstrated the strongest association (r = 0.998), followed by meaning in life (r = 0.562). Faith had a weaker, non-significant correlation (r = 0.333). **(B)** Group centroids representing the canonical discriminant scores for participants with and without ED. Individuals without ED had a positive centroid (+0.325), while those with ED had a negative centroid (–0.504), indicating significant group separation. *Note:* Wilks' Lambda = 0.858, *p* < 0.001.

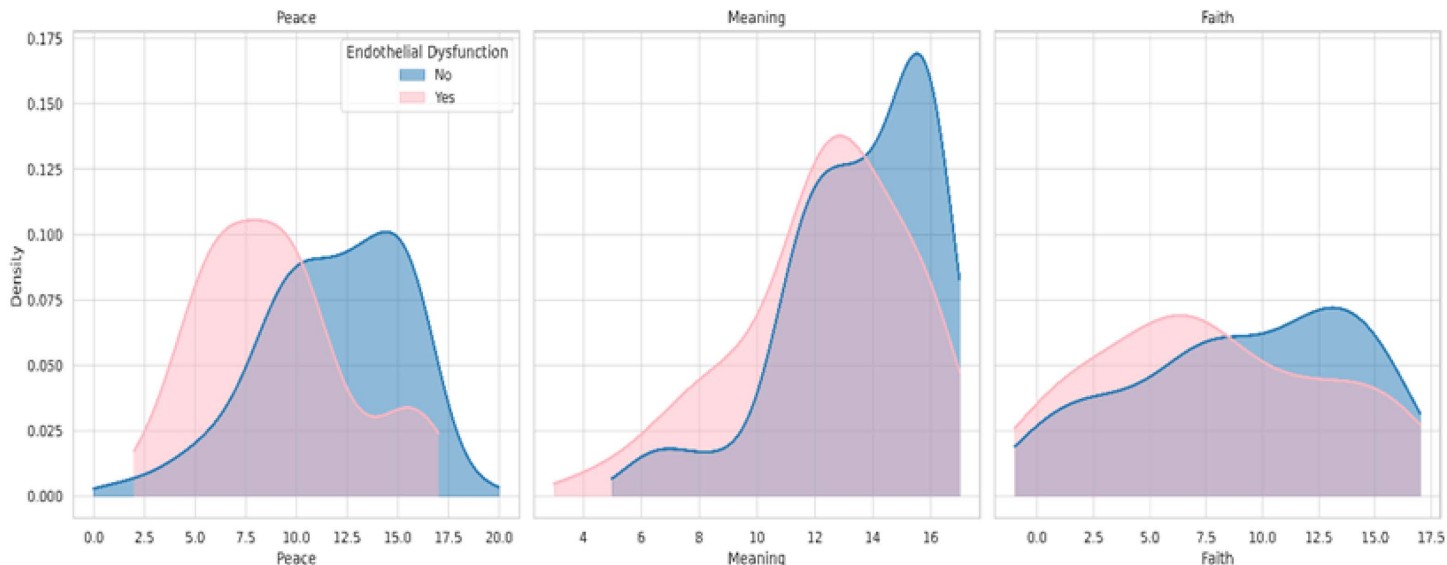

**Fig 3. Kernel density estimates for SWB dimensions by ED status.** Density plots comparing the distribution of scores on the Peace, Meaning in Life, and Faith subscales of the FACIT-Sp, stratified by presence (pink) or absence (blue) of ED. Participants without ED consistently showed higher density peaks across all subscales, particularly for inner peace and meaning, indicating a potential protective role of these dimensions in vascular health.

In addition to these general effects, discriminant analysis revealed which specific dimensions of spirituality are most relevant for vascular health. Inner peace and life purpose were the only dimensions that significantly discriminated between individuals with and without ED. Inner peace showed the highest discriminant power, suggesting that experiences of emotional tranquility and existential coherence may be central to maintaining endothelial homeostasis. In contrast, the faith dimension was not statistically significant, highlighting the importance of personal spiritual experiences over doctrinal or institutional religious practices in this context.

The greater discriminative power of peace and meaning—independent of faith—suggests that the potential vascular benefits associated with SWB may extend beyond institutional religiosity. This supports the universal relevance of existential dimensions of spirituality and their possible value in cardiovascular risk assessment across culturally diverse populations.

Our findings are further supported by physiological data [25,26]. While traditional risk factors such as age, sex, and physical activity did not reach significance in the regression model, BMI remained positively associated with ED. This reinforces the established link between adiposity and endothelial impairment, mediated by mechanisms such as systemic inflammation, oxidative stress, and insulin resistance [27]. The absence of association with age suggests that ED is not merely a function of chronological aging but also reflects psychosocial and behavioral influences.

The interplay between spirituality and physiological health is further illustrated by evidence linking SWB with reduced inflammatory markers. Prior studies have reported associations between SWB and lower levels of IL-6 and C-reactive protein [28,29] both key mediators of ED and atherogenesis. In this regard, our findings echo those of the FEEL study, which demonstrated that spirituality-based interventions can enhance flow-mediated dilation and lower blood pressure in hypertensive individuals [29]. Findings from the FEEL Study support the biological plausibility of our results, suggesting that spiritually oriented practices may benefit endothelial function. Likewise, the Jackson Heart Study [30] found that higher levels of religiosity and spirituality were associated with healthier behaviors, improved clinical indicators, and more favorable cardiovascular profiles, underscoring the role of spiritual engagement in culturally sensitive prevention strategies.

The robust discriminant value of inner peace and life purpose suggests that SWB may act through mechanisms related to emotional regulation and stress resilience. Inner peace may buffer the impact of psychosocial stressors by attenuating sympathetic activation and preserving nitric oxide bioavailability—critical for vascular tone and endothelial repair. The conceptual model proposed by Vos [31], linking meaning in life to improved cardiovascular outcomes, supports this hypothesis. Our data reinforce this model by demonstrating that meaning and peace are directly associated with vascular health in a clinically healthy Brazilian population.

Academic interest in spirituality and health has also grown in Brazil, with a recent national survey identifying 36 research groups focused on this topic, many addressing cardiovascular outcomes such as hypertension and coronary artery disease [32]. Predominantly based in public universities, these groups reflect a multidisciplinary and integrative approach to care, reinforcing the national relevance of research on spirituality and vascular health.

From a clinical perspective, our results have important implications for clinical practice. While current guidelines [33] emphasize physiological risk factors such as hypertension and dyslipidemia, the present study suggests that incorporating SWB into risk assessments could provide a more holistic and predictive approach. Spirituality assessment—alongside conventional measures—may help identify individuals at higher risk for subclinical vascular damage. Even in healthcare systems where spiritual health is officially encouraged—as in the UK—many physicians still report discomfort in addressing it during consultations. A recent study showed that only 50% of general practitioners felt confident discussing spirituality with patients, although most endorsed the usefulness of structured tools to guide such conversations [34]. Studies have identified common reasons for this gap, including lack of formal training, limited time during consultations, uncertainty about how to address spiritual issues appropriately, and fear of overstepping professional boundaries [35].

Moreover, interventions aimed at enhancing inner peace and existential meaning, such as mindfulness practices, cognitive-behavioral strategies, and meaning-centered therapies, could complement traditional cardiovascular care. These

findings provide new insights into the integration of psychosocial dimensions within preventive cardiology, particularly for asymptomatic individuals who may benefit from non-pharmacological strategies to support vascular function.

This study presents strengths that enhance the validity and applicability of its findings. These include the use of internationally validated instruments, strict exclusion criteria ensuring a clinically healthy sample, and the integration of psychological and spiritual variables in the assessment of cardiovascular risk. Together, these factors contribute to the methodological rigor and translational relevance of the study.

Limitations should be acknowledged. The cross-sectional nature of the study precludes causal inferences, and longitudinal data are needed to confirm whether increases in SWB lead to sustained improvements in endothelial function. Additionally, reliance on self-report measures may introduce bias, and physiological mechanisms were inferred but not directly assessed. The relatively small, exclusively Brazilian sample may limit the generalizability of our findings. Post hoc power analysis indicated adequate power for the primary model but reduced power after excluding participants with anxiety or depression. The logistic regression model explained 28.8% of the variance in ED, leaving substantial unexplained variability likely related to unmeasured factors such as inflammatory biomarkers, autonomic function, or lifestyle variables. After excluding mental health comorbidities, the explained variance dropped to 9.7%, despite similar classification accuracy (75,3%), underscoring their substantial contribution to endothelial function and suggesting attenuation of associations in populations without these conditions. Future studies should explore the potential mediating role of SWB in cardiovascular health using larger, more diverse samples, longitudinal designs, objective biomarkers, and interventional trials targeting both modifiable and non-modifiable risk factors.

## Conclusion

This study reveals a significant association between higher levels of SWB—especially inner peace and a sense of purpose—and more favorable endothelial function in clinically healthy adults. Findings suggest that psychosocial factors play a meaningful role in modulating vascular health, potentially mitigating stress-related endothelial impairment. A holistic approach integrating spiritual and physiological factors may enhance cardiovascular prevention strategies.

## Acknowledgments

The authors would like to thank the research team of the Núcleo de Pesquisas em Espiritualidade (NESPE) at the University Hospital Gaffrée e Guinle (UNIRIO) for their valuable assistance in participant recruitment and data collection. We are also grateful to the clinical staff who supported the cardiovascular assessments and to all study participants for their generous collaboration.

## Author contributions

**Conceptualization:** Andre Casarsa, Pedro Bastos de Medeiros, Julio Cesar Tolentino, Maria de Fatima Martins Gil Dias, Sergio Luis Schmidt.

**Data curation:** Andre Casarsa, Pedro Bastos de Medeiros, Julio Cesar Tolentino, Isadora de Sa Guimaraes, Kelen Carolina Silva Cruz, Leticia Silva Flor dos Santos, Matheus Nakazato Tinoco, Maria de Fatima Martins Gil Dias, Ana Lucia Taboada Gjorup, Sergio Luis Schmidt.

**Formal analysis:** Andre Casarsa, Pedro Bastos de Medeiros, Julio Cesar Tolentino, Ana Lucia Taboada Gjorup, Sergio Luis Schmidt.

**Investigation:** Andre Casarsa, Pedro Bastos de Medeiros, Julio Cesar Tolentino, Isadora de Sa Guimaraes, Kelen Carolina Silva Cruz, Leticia Silva Flor dos Santos, Matheus Nakazato Tinoco, Maria de Fatima Martins Gil Dias, Ana Lucia Taboada Gjorup, Sergio Luis Schmidt.

**Methodology:** Andre Casarsa, Pedro Bastos de Medeiros, Julio Cesar Tolentino, Sergio Luis Schmidt.

**Project administration:** Andre Casarsa, Sergio Luis Schmidt.

**Supervision:** Andre Casarsa, Sergio Luis Schmidt.

**Validation:** Andre Casarsa, Pedro Bastos de Medeiros, Sergio Luis Schmidt.

**Visualization:** Pedro Bastos de Medeiros, Sergio Luis Schmidt.

**Writing – original draft:** Andre Casarsa, Pedro Bastos de Medeiros, Julio Cesar Tolentino, Ana Lucia Taboada Gjorup, Sergio Luis Schmidt.

**Writing – review & editing:** Andre Casarsa, Pedro Bastos de Medeiros, Julio Cesar Tolentino, Sergio Luis Schmidt.

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
