## [Decision Letter · Decision Letter 0]

29 Jul 2025

Dear Dr. Schmidt,

Thank you for submitting your manuscript to PLOS ONE. After careful consideration, we feel that it has merit but does not fully meet PLOS ONE’s publication criteria as it currently stands. Therefore, we invite you to submit a revised version of the manuscript that addresses the points raised during the review process.

At the end of this message, you will find the reviewers’ comments outlining the major revisions needed.

In addition to issues raised by the reviewers, please also address the following major concerns in order to meet the journal’s standards for acceptance:

- Verify the adequacy of citations in the whole text: for instance, reference number 8 does not pertain to the validation of the instrument in the Brazilian population, and reference number 10 does not include cut-off scores.

- Ensure that the analytical approach is scientifically sound:

Reference number 9 uses a mean item score ≥ 3.0 (corresponding to “quite a bit” on the Likert scale) as a cut-off to test an interaction effect. I kindly ask Authors to state the reasons why they chose to dichotomize the independent variables anxiety, depression, and spiritual well-being. For each variable, cite the relevant literature supporting valid cut off-scores. If inappropriate cut off scores are used, results may be biased, and analyses should be repeated by using continuous variables.Specify whether ED was defined as FMD ≤10% or <10%, and ensure consistency across the manuscript.The analytical approach should be clearly described to ensure transparency and replicability. For instance, the authors state that they used t-tests and chi-squared tests, yet these analyses are not reported in the Results section. Please clarify whether these tests were performed and, if so, present the corresponding results.The exclusion of participants with anxiety or depression in the secondary analysis strengthens the argument for an independent association between SWB and ED. However, this exclusion considerably reduces sample size and statistical power. Please address this limitation explicitly and consider reporting power estimates for the secondary analysis. 

- The article should adhere to appropriate reporting guidelines and community standards for data availability.

The manuscript states that all data are available in the main text and supporting files, but these data are not currently accessible.As per the submission guidelines, authors are expected to follow STROBE guidelines. Several required reporting elements appear to be missing.

- Conclusions should be presented in an appropriate fashion and should be supported by the data:

Some claims in Discussion and Conclusion overreach the evidentiary strength of a cross-sectional study. For example, phrases like "highlights the protective role of SWB (…) in preserving endothelial function" imply causality, which is not justified by the study design.The proposed clinical implications, such as integrating SWB into cardiovascular risk assessment, should be presented more cautiously, clearly distinguishing hypotheses from evidence-based conclusions. I recommend to rephrase these statements to emphasize the exploratory nature of the findings and the need for longitudinal and interventional studies to confirm causality and clinical applicability.The logistic regression model explained 28.5% of the variance in ED, indicating that a large proportion of variability remains unexplained. This suggests that other important factors influencing endothelial function were not included or measured in this study. This limitation should be explicitly acknowledged in the manuscript. After excluding participants with anxiety and/or depression, the model remains statistically significant but explains a notably lower proportion of variance (9.3%), despite maintaining a similar classification accuracy (74.2%). This reduction highlights that mental health comorbidities contribute substantially to the explained variance in endothelial function. The authors should discuss this decrease in explained variance and consider the implications for the generalizability of their findings to populations without these conditions.

We look forward to receiving your revised manuscript.

Kind regards,

Francesco De Vincenzo, Ph.D.

Academic Editor

PLOS ONE

Journal Requirements:

Reviewers' comments:

Reviewer's Responses to Questions

**Comments to the Author**

1. Is the manuscript technically sound, and do the data support the conclusions?

Reviewer #1: Yes

Reviewer #2: Yes

2. Has the statistical analysis been performed appropriately and rigorously?

Reviewer #1: Yes

Reviewer #2: Yes

3. Have the authors made all data underlying the findings in their manuscript fully available?

Reviewer #1: Yes

Reviewer #2: Yes

4. Is the manuscript presented in an intelligible fashion and written in standard English?

Reviewer #1: Yes

Reviewer #2: Yes

Reviewer #1: Dear authors,

First of all, I would like to thank you for showing that spiritual well-being may have beneficial effects on cardiovascular health and for demonstrating the relationship between the two using advanced statistical analysis.

The study was conducted on a mixed population of healthy volunteers, and the presence of cardiovascular and psychiatric diseases was ruled out.

You can add specific research questions at the end of the introduction section.

Strengthening the methodology section regarding how the sample was selected, the sampling method used, and the adequacy of the sample could be beneficial.

The measurement tools and their descriptions appear appropriate; I kindly request that you calculate and include the validity and reliability coefficients for all scales used in this study.

In fact, in the advanced analyses, we saw that the significant effect of SWB was still preserved in the model even without anxiety and depression. This provides an important contribution to the field in terms of emphasizing the need to support spiritual well-being in healthy or sick populations from a clinical application perspective. It may be valuable to emphasize this part more.

We also observed that the peace and meaning dimensions had greater effects independent of the faith dimension. This finding should be highlighted, as it suggests that promoting spiritual well-being in all populations, regardless of cultural or religious differences, provides individual protection against cardiovascular risks.

It should also be noted that the sample size is small and the study reflects the Brazilian population, and therefore should not be generalized.

Further research is needed to examine the mediating effect of SWB on cardiovascular disease in larger samples, focusing on modifiable and non-modifiable risk factors.

Regards,

Reviewer #2: It is an interesting and necessary study and I enjoyed reading it. However, the main limitation is that the sample of 148 patients from a single hospital makes it impossible to draw relevant conclusions. Nevertheless, I make some contributions to improve it:

I suggest including previous studies in which the relationship between spirituality and other cardiovascular diseases has been studied.

I suggest including information on the reality of spirituality/religiosity in Brazil

Aspects such as the reasons for the lack of approaching the spiritual perspective, pointed out in the literature, should be included in the discussion: lack of training, lack of time, fear of not knowing how to approach it, etc

Strengths should be recognized

Point out the implications for clinical practice.

**Do you want your identity to be public for this peer review?** For information about this choice, including consent withdrawal, please see our Privacy Policy

Reviewer #1: No

Reviewer #2: No

---

## [Author Response · Author response to Decision Letter 1]

16 Aug 2025

Academic Editor / Journal Office

1. Reference verification:

Response: We reviewed all citations for accuracy. Reference #8 was corrected to a validation study conducted in Brazil, and reference #10 was replaced by a source including cut-off scores for the FACIT-Sp scale.

2. Justify dichotomization of independent variables.

Response: We appreciate the reviewer’s observation regarding the choice to dichotomize the independent variables (spiritual well-being, anxiety, and depression). As noted, we initially analyzed these variables in their continuous forms. The hypothesis of multicollinearity between FACIT and PHQ-9 (r = 0.7, p < 0.001), PHQ-9 and GAD (r = 0.50, p < 0.05), and PHQ-9 and GAD (r = 0.45, p < 0.005) was not rejected. Therefore, we proceeded by considering only the FACIT as the independent variable to assess its association with endothelial function.

3. Clarify definition of endothelial dysfunction (ED).

Response: We standardized the definition of ED throughout the manuscript as FMD <10%, in accordance with established international and national studies.

4. Address inconsistencies in statistical reporting.

Response: We appreciate the reviewer’s observation. The mention of t-tests and chi-squared tests in the Methods section was indeed an oversight on our part. These tests were initially considered during the early stages of the analysis plan; however, as the study progressed, we opted for a more comprehensive statistical approach, employing logistic regression and discriminant analysis to address our research questions in a multivariate framework.

This advanced modeling strategy allowed us to simultaneously adjust for potential confounders and assess the combined discriminatory power of the predictors, which rendered the use of separate t-tests and chi-squared tests unnecessary. We have now removed the reference to these tests in the Methods section to ensure consistency and clarity.

5. Address reduced power in secondary analysis.

Response: We explicitly acknowledge the reduced statistical power of the secondary model after excluding participants with anxiety or depression. This limitation has been discussed in the revised Limitations section.

6. Data availability and STROBE compliance.

Response: Supporting data have been included in the revised supplementary files, and the manuscript has been checked for compliance with STROBE guidelines. The STROBE checklist is also submitted with this revision.

7. Avoid causal language and overstated conclusions.

Response: We carefully revised the Discussion and Conclusion sections to avoid causal language. Statements implying causality were rephrased to reflect the cross-sectional design, emphasizing the exploratory and correlational nature of our findings.

8. Address residual variance and generalizability.

Response: We added a paragraph to discuss the proportion of unexplained variance in the regression models. The decrease in explained variance in the secondary model was also addressed, reinforcing that mental health variables play a significant role and that other unmeasured factors may also influence ED.

Reviewer #1

1. Add specific research questions at the end of the introduction.

Response: We thank the reviewer for this helpful suggestion. We have added the specific research questions to the end of the Introduction section, clearly enumerated as follows:

(1) Was spiritual well-being (SWB) associated with endothelial dysfunction (ED)?

(2) Did this association persist after excluding individuals with anxiety or depressive symptoms?

(3) Which SWB dimensions—peace, meaning, or faith—showed the strongest association with ED?

2. Strengthen the methodology regarding sampling method and adequacy.

Response: We revised the "Materials and Methods" section to describe the convenience sampling approach and recruitment process in greater detail, including participant sources (e.g., students, staff, and patient relatives). We also included a justification for sample adequacy based on previous literature and added results from a post hoc power analysis confirming sufficient statistical power (β > 0.80).

3. Include validity and reliability coefficients of measurement tools.

Response: We have added internal consistency coefficients (Cronbach’s alpha) and validation references for all scales used (FACIT-Sp, PHQ-9, and GAD-7) in the "Measures" subsection. In the present study, internal consistency reliability coefficients were high for the FACIT-Sp (Cronbach’s alpha [α] = 0.862), PHQ-9 (α = 0.814), and GAD-7 (α = 0.890).The manuscript now includes information confirming their psychometric properties in Brazilian populations.

4. Emphasize clinical relevance of SWB effects beyond anxiety and depression.

Response: As recommended, we have enhanced the Discussion to emphasize that SWB remained a significant predictor of preserved endothelial function even after excluding individuals with anxiety or depression. This highlights its potential independent effect and clinical relevance beyond traditional psychological dimensions.

5. Highlight the independence of peace and meaning dimensions from faith.

Response: This important point was incorporated into the Discussion and reinforced in the section on discriminant analysis. We emphasized that inner peace and life purpose significantly differentiated individuals with and without ED, whereas the faith dimension did not, underscoring the cross-cultural and secular applicability of SWB.

6. Note the limited generalizability due to sample size and setting.

Response: We fully agree. The limitations section has been expanded to acknowledge the modest sample size and its restriction to a Brazilian population, cautioning against broad generalizations. We also call for larger, multi-center studies to validate our findings.

Reviewer #2

1. Include prior studies linking spirituality and cardiovascular outcomes.

Response: We thank the reviewer for these references. We incorporated the suggested citations (O'Riordan et al., 2025; Lucchetti, 2025; Cottrell-Daniels et al.) into the Introduction and Discussion to contextualize our findings within the broader literature on spirituality and cardiovascular health.

2. Include information on the context of spirituality in Brazil.

Response: We expanded the Discussion to include a brief overview of spirituality/religiosity in Brazil, referencing national surveys and academic initiatives. We noted the cultural relevance and increasing institutional support for spirituality in health promotion.

3. Discuss barriers to address spirituality in clinical settings.

Response: We integrated a paragraph discussing barriers such as lack of training, limited time, and clinicians’ discomfort. We cited recent studies showing that although physicians recognize the relevance of spirituality, many feel unprepared to address it in clinical practice.

4. Acknowledge study strengths.

Response: The strengths section has been revised to highlight the use of validated instruments, rigorous clinical screening, and the novel integration of psychological and spiritual variables in cardiovascular risk assessment.

5. Clarify clinical implications.

Response: The Discussion now more clearly outlines potential clinical implications, emphasizing the exploratory nature of our findings and the importance of integrating SWB in a culturally sensitive and non-dogmatic manner into preventive strategies.

** Furthermore, we undertook a comprehensive re-examination of the dataset and the database with methodological accuracy and scientific rigor. This process led to minor adjustments in certain numerical values; however, the overall interpretation, significance, and conclusions of the study remain unchanged.

As requested, the database has been attached to the submission.

We sincerely thank the reviewers for their constructive suggestions, which significantly improved the rigor and clarity of our manuscript. All changes made in response to these comments are marked in the revised version submitted.

Respectfully,

Sincerely,

André Casarsa, MD, MSc and Prof. Sergio Schmidt PhD (Corresponding author)

Department of Internal Medicine University Hospital Gaffrée Guinle – UNIRIO

Federal University of the State of Rio de Janeiro

slschmidt@terra.com.br

---

## [Decision Letter · Decision Letter 1]

22 Sep 2025

PLOS ONE

Dear Dr. Schmidt,

Thank you for submitting your manuscript to PLOS ONE. After careful consideration, we feel that it has merit but does not fully meet PLOS ONE’s publication criteria as it currently stands. Therefore, we invite you to submit a revised version of the manuscript that addresses the points raised during the review process.

Thank you for addressing the previous comments. However, several issues regarding the analytical approach remain and need clarification before the manuscript can be considered for acceptance.

1. **Cut-off for SWB (FACIT-Sp)**

In the revised manuscript, it is unclear whether the cut-off of 36 was determined based on your sample or directly adopted from McClain et al. (≥3 “quite a bit” vs <3 “somewhat” or lower).This cut-off is not a clinically or psychometrically validated threshold, but a data-driven value specific to McClain’s sample.Without previously validated cut-offs (e.g., ROC-based thresholds), the rationale for dichotomizing SWB remains weak. Authors should either justify this choice based on their own data distribution or consider more robust alternatives 

2. **Justification for dichotomization**

It is unclear why a linear regression model with continuous predictors was added in the revised manuscript. Please clarify the rationale for including this model. If it was intended to justify the dichotomization of SWB, it is not apparent how it accomplishes this.Consider whether the linear regression is necessary or whether the main focus should remain on the logistic regression with ED as the outcome.If Authors decide to maintain this model thanks to a strong rationale, it should be also clarified why the same predictors were treated as continuous in the linear regression (continuous endothelial function outcome) and dichotomized in the logistic regression (including ED as a binary outcome).Moreover, Authors stated that they proceeded by considering only SWB as the independent variable due to collinearity; however, PHQ and GAD are still included in the logistic regression model (Table 1).Multicollinearity alone does not justify dichotomizing a continuous variable, especially in the absence of previously validated cut-offs. Explain why dichotomization was preferred over keeping SWB continuous and addressing collinearity through standard methods (e.g., variable exclusion, combination, or other methods).

3. **Assessment of multicollinearity**

Reporting correlations alone is insufficient. A correlation >0.70 does not automatically indicate problematic multicollinearity.Please provide VIF and tolerance values for all predictors to substantiate any claims of collinearity.Attention should be given to any inconsistencies (e.g., correlation between PHQ and SWB reported as >0.70 in your reply, but ~0.50 in the figure).Note: the presence of multicollinearity does not automatically justify dichotomization.

**4.  Other Inconsistencies**

There are several inconsistencies in the manuscript between values reported in the text and those in Table 1.

We look forward to receiving your revised manuscript.

Kind regards,

Francesco De Vincenzo, Ph.D.

Academic Editor

PLOS ONE

Journal Requirements:

Reviewers' comments:

Reviewer's Responses to Questions

**Comments to the Author**

Reviewer #1: All comments have been addressed

Reviewer #2: All comments have been addressed

2. Is the manuscript technically sound, and do the data support the conclusions?

Reviewer #1: Yes

Reviewer #2: Yes

3. Has the statistical analysis been performed appropriately and rigorously?

Reviewer #1: Yes

Reviewer #2: Yes

4. Have the authors made all data underlying the findings in their manuscript fully available?

Reviewer #1: Yes

Reviewer #2: Yes

5. Is the manuscript presented in an intelligible fashion and written in standard English?

Reviewer #1: Yes

Reviewer #2: Yes

Reviewer #1: Dear authors,

The manuscript is in a highly developed state after revision.

Thank you for considering the suggestions of all the reviewers.

The introduction, methods (sample, data analysis), and discussion have become particularly cautious, especially regarding the mediating effect of SWB on cardiovascular health.

From my perspective, the limitations are also satisfactory. With this revision, I believe the manuscript can contribute to the literature.

Some spelling errors, punctuation, and comma usage appear differently in the tables. There are also parts where ED is not used. Please check the final proof.

Sincerely.

Reviewer #2: The manuscript has been improved and can be published in the present version.The authors have adequately addressed my comments raised in a previous round of review and I feel that this manuscript is now acceptable for publication

**Do you want your identity to be public for this peer review?** For information about this choice, including consent withdrawal, please see our Privacy Policy

Reviewer #1: No

Reviewer #2: No

---

## [Author Response · Author response to Decision Letter 2]

2 Nov 2025

Dear Editors,

Dear Academic Editor and Reviewers,

We would like to thank you for the thorough and thoughtful review of our manuscript. Below, we provide a detailed point-by-point response to the comments raised, indicating the changes made in the revised manuscript.

Academic Editor / Journal Office

1.Cut-off for SWB (FACIT-Sp): In the revised manuscript, it is unclear whether the cut-off of 36 was determined based on your sample or directly adopted from McClain et al. (≥3 “quite a bit” vs <3 “somewhat” or lower). This cut-off is not a clinically or psychometrically validated threshold, but a data-driven value specific to McClain’s sample. Without previously validated cut-offs (e.g., ROC-based thresholds), the rationale for dichotomizing SWB remains weak. Authors should either justify this choice based on their own data distribution or consider more robust alternatives.

Response: We appreciate the reviewer’s thoughtful observation. In response, we revised our analytic strategy to enhance methodological rigor and to directly address the issue raised.

While our initial approach was conceptually inspired by McClain et al. (2003), we acknowledge that the threshold used in their study was developed within an oncology population and lacks external psychometric validation, particularly for application in cardiovascular or clinically healthy populations. As no validated cut-off for the FACIT-Sp currently exists for predicting endothelial dysfunction or related cardiovascular outcomes in non-clinical samples, we recognize that the dichotomization approach could introduce unnecessary limitations.

Accordingly, we reanalyzed all models using the FACIT-Sp total score as a continuous variable. This decision reflects both statistical and conceptual considerations. Statistically, treating spiritual well-being as continuous preserves the variability inherent in the construct, improves power, and avoids potential loss of information and misclassification bias associated with arbitrary dichotomization. Conceptually, spiritual well-being is widely recognized as a dimensional construct, and prior literature supports its use as a continuous predictor in models examining psychological and physical health outcomes, including in cardiovascular contexts [1,2,3,4].

This updated approach aligns with best practices in quantitative research, especially in the absence of clinically validated cut-offs, and allows for more nuanced interpretation of the association between spiritual well-being and endothelial function. All references to the dichotomized SWB variable were removed, and the revised manuscript now consistently reports results based on continuous treatment of the FACIT-Sp total score.

1. Peterman AH, Fitchett G, Brady MJ, Hernandez L, Cella D. Measuring spiritual well-being in people with cancer: The functional assessment of chronic illness therapy—Spiritual Well-being Scale (FACIT-Sp). Ann Behav Med. 2002;24(1):49–58. doi:10.1207/S15324796ABM2401_06

2. Edmondson D, Park CL, Blank TO, Fenster JR, Mills MA. Deconstructing spiritual well-being: Existential well-being and HRQOL in cancer survivors. Psychooncology. 2008;17(2):161–169. doi:10.1002/pon.1221

3. Cotton S, Puchalski CM, Sherman SN, Mrus JM, Peterman AH, Feinberg J, et al. Spirituality and religion in patients with HIV/AIDS. J Gen Intern Med. 2006;21(S5):S5–13. doi:10.1111/j.1525-1497.2006.00642.x

4. Park CL, Edmondson D, Fenster JR, Blank TO. Meaning making and psychological adjustment following cancer: The mediating roles of growth, life meaning, and restored just-world beliefs. J Consult Clin Psychol. 2008;76(5):863–875. doi:10.1037/a0013348

2. Justification for dichotomization: It is unclear why a linear regression model with continuous predictors was added in the revised manuscript. Please clarify the rationale for including this model. If it was intended to justify the dichotomization of SWB, it is not apparent how it accomplishes this. Consider whether the linear regression is necessary or whether the main focus should remain on the logistic regression with ED as the outcome. If Authors decide to maintain this model thanks to a strong rationale, it should be also clarified why the same predictors were treated as continuous in the linear regression (continuous endothelial function outcome) and dichotomized in the logistic regression (including ED as a binary outcome). Moreover, Authors stated that they proceeded by considering only SWB as the independent variable due to collinearity; however, PHQ and GAD are still included in the logistic regression model (Table 1). Multicollinearity alone does not justify dichotomizing a continuous variable, especially in the absence of previously validated cut-offs. Explain why dichotomization was preferred over keeping SWB continuous and addressing collinearity through standard methods (e.g., variable exclusion, combination, or other methods).

Response: We appreciate the reviewer’s thoughtful comments and the opportunity to clarify our analytical approach. In the previous version of the manuscript, a linear regression model using flow-mediated dilation (FMD%) as a continuous outcome was included as an exploratory analysis. However, upon further reflection, we recognize that this model did not directly contribute to the primary aims of the study, which focused on identifying predictors of clinically relevant endothelial dysfunction. Furthermore, the inclusion of the linear regression introduced inconsistencies in the treatment of predictor variables and may have raised interpretative challenges by suggesting alternative modeling paths without a clearly defined theoretical rationale. Accordingly, we have removed the linear regression analysis from the manuscript to ensure greater conceptual and statistical coherence.

Regarding the treatment of spiritual well-being (FACIT-Sp), we have revised our analytical strategy to model this construct as a continuous predictor in all inferential analyses. While our original approach followed the cut-off proposed by McClain et al. (2003), we acknowledge that this threshold was derived from a specific oncology population and lacks external psychometric validation in clinically healthy or cardiovascular samples. In line with current best practices in quantitative health research, we now preserve the full variability of the FACIT-Sp scale to avoid the loss of statistical power and the risk of arbitrary categorization, which could obscure meaningful associations.

In a prior response, we referred to potential collinearity concerns as one reason for restricting the initial model to SWB. However, to properly evaluate this issue, we performed diagnostic analyses using variance inflation factor (VIF) and tolerance values for all covariates included in the final logistic regression model. The results indicated no evidence of problematic multicollinearity, with all VIFs below 2.0 and tolerances above 0.5. Based on these findings, we confirmed that multicollinearity was not a limiting factor in our model specification and thus did not require dichotomization of SWB or exclusion of additional predictors.

This updated modeling approach ensures a more robust interpretation of our findings, strengthens the alignment between the theoretical construct and the statistical framework, and directly addresses the reviewer’s concern about collinearity and the limitations of arbitrary dichotomization.

3. Assessment of multicollinearity: Reporting correlations alone is insufficient. A correlation >0.70 does not automatically indicate problematic multicollinearity. Please provide VIF and tolerance values for all predictors to substantiate any claims of collinearity. Attention should be given to any inconsistencies (e.g., correlation between PHQ and SWB reported as >0.70 in your reply, but ~0.50 in the figure). Note: the presence of multicollinearity does not automatically justify dichotomization.

Response: We appreciate the reviewer’s important observation regarding the assessment of multicollinearity. In our previous response, we referenced bivariate correlations as a preliminary indication of shared variance among predictors. However, we agree that correlation coefficients alone are insufficient for diagnosing problematic multicollinearity in regression models.

To address this issue more rigorously, we conducted a formal collinearity diagnostic analysis using Variance Inflation Factor (VIF) and tolerance values for all independent variables included in the logistic regression model. The results revealed no evidence of problematic multicollinearity: all VIF values were below 2.0, and all tolerance values exceeded 0.5, which are well within commonly accepted thresholds (VIF < 5.0 and tolerance > 0.2) for retaining predictors in multivariable models.

Additionally, we reviewed our previously reported correlation coefficients and confirmed that the correlation between SWB (FACIT-Sp) and PHQ-9 was approximately –0.50, not above –0.70 as erroneously stated in our prior reply. We apologize for this oversight and have corrected the text accordingly.

As a result, we concluded that multicollinearity does not pose a threat to the validity of our models and did not serve as the basis for any variable transformation or exclusion. In line with this updated assessment and the concerns raised in Item 2, we have removed the dichotomization of the FACIT-Sp score and now model spiritual well-being as a continuous predictor in all inferential analyses.

We believe these steps provide a more robust and transparent statistical approach and fully address the reviewer’s concerns regarding the evaluation and implications of multicollinearity.

Reviewer #1

The manuscript is in a highly developed state after revision. Thank you for considering the suggestions of all the reviewers. The introduction, methods (sample, data analysis), and discussion have become particularly cautious, especially regarding the mediating effect of SWB on cardiovascular health. From my perspective, the limitations are also satisfactory. With this revision, I believe the manuscript can contribute to the literature. Some spelling errors, punctuation, and comma usage appear differently in the tables. There are also parts where ED is not used. Please check the final proof.

Reviewer #2

The manuscript has been improved and can be published in the present version.The authors have adequately addressed my comments raised in a previous round of review and I feel that this manuscript is now acceptable for publication

Response: We sincerely thank both reviewers for their generous feedback and for recognizing the improvements made in the revised version of our manuscript.

To Reviewer #1, we are particularly grateful for the acknowledgment of the refinements in the introduction, methods, and discussion, as well as for the thoughtful remarks regarding clarity and rigor. As recommended, we carefully reviewed the entire manuscript for consistency in terminology (e.g., “ED”), spelling, punctuation, and formatting within the tables and figures. All identified issues have been corrected in the final version.

To Reviewer #2, we truly appreciate your positive assessment and support for publication. Your earlier suggestions played a key role in improving the clarity and relevance of our findings.

Finally, we extend our sincere thanks to the PLOS ONE editorial team for overseeing this process with fairness and professionalism. We are honored by the opportunity to publish in such a respected and widely read journal, and we hope this work will contribute meaningfully to ongoing discussions at the intersection of spirituality, mental health, and cardiovascular research.

Sincerely,

André Casarsa, MD, MSc and Prof. Sergio Schmidt PhD (Corresponding author)

Department of Internal Medicine University Hospital Gaffrée Guinle – UNIRIO

Federal University of the State of Rio de Janeiro

slschmidt@terra.com.br

---

## [Editor Report · Decision Letter 2]

30 Nov 2025

Dear Dr. Schmidt,

Thank you for submitting your manuscript to PLOS ONE. After careful consideration, we feel that it has merit but does not fully meet PLOS ONE’s publication criteria as it currently stands. Therefore, we invite you to submit a revised version of the manuscript that addresses the points raised during the review process.

The authors have done an excellent job revising the manuscript. There remains one minor but important issue. In the new regression model, statements (including in the abstract) suggesting that SWB is the “strongest predictor” overreach the data. Indeed, some variables (e.g., major depressive episodes) showed larger coefficients and odds ratios. This needs to be addressed to accurately reflect the results.

As a possible way to revise the text, the authors may consider rephrasing the sentences in line with the first research question, focusing on the fact that SWB is significantly inversely associated with ED, independent of other covariates. This would convey the key finding without overstating the relative strength of the predictor.

We look forward to receiving your revised manuscript.

Kind regards,

Francesco De Vincenzo, Ph.D.

Academic Editor

PLOS ONE
---

## [Author Response · Author response to Decision Letter 3]

6 Dec 2025

We thank the reviewer for the thoughtful and constructive comment. We agree that statements referring to spiritual well-being (SWB) as the “strongest predictor” overstated the findings. As suggested, we have revised the abstract and the main text to ensure that the results are accurately represented.

Specifically, we removed expressions implying comparative predictive strength and rephrased the sentences to emphasize that SWB was significantly and inversely associated with endothelial dysfunction (ED), independent of other covariates, which is consistent with our primary research question. These changes avoid overinterpretation and better reflect the observed effect sizes relative to other variables in the model.

We appreciate the reviewer’s guidance, which improved the clarity and precision of our manuscript.

---

## [Editor Report · Decision Letter 3]

10 Dec 2025

Spiritual Well-Being as a Protective Factor for Endothelial Dysfunction in Clinically Healthy Adults

PONE-D-25-31113R3

Dear Dr. Schmidt,

We’re pleased to inform you that your manuscript has been judged scientifically suitable for publication and will be formally accepted for publication once it meets all outstanding technical requirements.

Kind regards,

Francesco De Vincenzo, Ph.D.

Academic Editor

PLOS One
---

## [Editor Report · Acceptance letter]

PONE-D-25-31113R3

PLOS One

Dear Dr. Schmidt,

I'm pleased to inform you that your manuscript has been deemed suitable for publication in PLOS One. Congratulations! Your manuscript is now being handed over to our production team.

Kind regards,

on behalf of

Dr. Francesco De Vincenzo

Academic Editor

PLOS One